# Significance of descriptive symptoms and signs and clinical parameters as predictors of neuropathic cancer pain

Sun Kyung Baek[1], Sang Won Shin[2]*, Su-Jin Koh[3], Jung Han Kim[4], Hyo Jung Kim[5], Byoung Yong Shim[6], Seok Yun Kang[7], Sang Byung Bae[8], Hwan Jung Yun[9], Sun Jin Sym[10], Hye Sook Han[11], Ha Yeong Gil[12]

1 Internal Medicine, Kyung Hee University Medical Center, Seoul, South Korea, 2 Internal Medicine, Korea University College of Medicine, Seoul, South Korea, 3 Department of Hematology-Oncology, Ulsan University Hospital, Ulsan University College of Medicine, Ulsan, South Korea, 4 Internal Medicine, Kangnam Sacred-Heart Hospital, Hallym University College of Medicine, Seoul, South Korea, 5 Internal Medicine, Hallym University Sacred Heart Hospital Anyang, Anyang, South Korea, 6 Internal Medicine, St. Vincent's Hospital, The Catholic University of Korea, Suwon, South Korea, 7 Department of Hematology-Oncology, Ajou University School of Medicine, Suwon, South Korea, 8 Internal Medicine, Soonchunhyang University College of Medicine, Cheonan, South Korea, 9 Internal Medicine, Chungnam National University College of Medicine, Daejeon, South Korea, 10 Internal Medicine, Gachon University Gil Medical Center, Incheon, South Korea, 11 Department of Internal Medicine, Chungbuk National University College of Medicine, Cheongju, South Korea, 12 Medical Affairs, Internal Medicine, Pfizer Pharmaceuticals Korea Ltd., Seoul, South Korea

* shinsw@kumc.or.kr

**Data Availability Statement:** Data underlying the study cannot be made publicly available due to ethical restrictions on personal data imposed by the central IRB of Korean Cancer study group and the IRB committees of the hospitals involved in the

## Abstract

### Purpose

Evaluation of symptoms and signs for the management of neuropathic cancer pain (NCP) is challenging. This study aimed to identify clinical predictors of NCP and symptoms and signs most relevant of those in Korean patients.

### Methods

This nationwide, descriptive, cross-sectional, multicenter, observational study included 2,003 cancer patients aged ≥20 years who reported a visual analog scale (VAS) score ≥1 for pain and provided informed consent for participation. The Douleur Neuropathic (DN4) questionnaire (score ≥4) was used to determine symptoms and signs as well as the presence of NCP.

### Results

The prevalence of NCP was associated with age <65 years [OR, 1.57; 95% CI, 1.270–1.934], disease duration >6 months (OR, 1.57; 95% CI, 1.232–2.012), stage IV cancer (OR, 0.75; 95% CI, 0.593–0.955), history of chemotherapy (OR, 1.74; 95% CI, 1.225–2.472), and moderate-to-severe cancer pain (OR, 2.05; 95% CI, 1.671–2.524) after multivariate analysis. The most common descriptive symptoms of NCP were tingling, electric shock, and pins and needles. For NCP patients in the presence or absence of the clinical predictors, pins

study. The participants in the study did not consent to the public sharing of their data. To access the data, Interested researchers may contact kcsg@kcsg.org for more information.

**Funding:** We confirm that this study was sponsored by Pfizer Pharmaceuticals Korea Ltd. Ha Yeong Gil, the employee of Pfizer Pharmaceuticals Korea Ltd., contributed in the study design, data analysis, decision to publish, or preparation of the manuscript.

**Competing interests:** We declare that the authors have no other competing interests. This does not alter our adherence to all the PLOS ONE policies on sharing data and materials.

and needles (p = 0.001) and painful cold (p<0.001) symptoms were significantly frequent in patients with moderate-to-severe pain. Tingling, numbness, and touch hypoesthesia (p = 0.022, 0.033, 0.024, respectively) were more frequent in those with longer cancer duration and hyperesthesia (p = 0.024) was more frequent in young patients.

## Conclusion

Age <65 years, disease duration >6 months, stage IV cancer, history of chemotherapy, and moderate-to-severe cancer pain, were identified as predictors of NCP. Some symptoms and signs of NCP were associated with these predictors. Further studies are warranted on the pathogenesis and management of NCP with respect to the symptoms and signs, and factors associated with pain severity in Korean patients.

## Introduction

One of the most fearful symptoms of cancer is pain. Chronic cancer pain is defined as chronic pain caused by the primary cancer itself, metastasis or its treatment [1]. Despite a plethora of guidelines and recommendations, cancer pain is frequently under-treated with more than 50% of patients with cancer describing the intensity of cancer pain as moderate or severe [2]. One of the barriers to pain control is inadequate pain assessment which, ideally, should consider the intensity and underlying etiology and mechanism [3]. While nociceptive pain is sensitive to opioids, the responsiveness of neuropathic pain to opioids is weak [4]. This is particularly important for neuropathic cancer pain (NCP) as a combination of non-opioid analgesic drugs with conventional opioid analgesia is required for the optimal management of NCP [5].

The International Association for the Study of Pain (IASP) classification of chronic pain for International Classification of Diseases (ICD-11) identifies specific codes for chronic NCP (neuropathic pain caused by a tumor), as well as neuropathic pain caused by a primary tumor or metastases damaging or injuring the peripheral or central nervous system [6]. Neuropathic pain affects up to 40% of cancer patients and is associated with increased pain intensity and decreased quality of life (QOL) [7–9]. Previously, we also reported the incidence rate of NCP in cancer patients in Korea and the relevance that NCP deteriorates patients' QOL [10]. For clinicians, the identification of NCP is dependent on the careful evaluation of the symptoms and signs. However, in daily clinical practice, identifying NCP can be difficult or sometimes easily neglected due to the nature of symptoms and NCP may be considered to be non-cancer associated pain [11,12]. To overcome this problem and to assist in the early detection of NCP, the present study evaluated clinical predictors of NCP and descriptive symptoms and signs associated with NCP among cancer patients in Korea.

## Materials and methods

### Data collection

This nationwide, cross-sectional, descriptive, non-interventional, multicenter study was conducted from February 2013 to March 2014 in 28 hospitals in Korea, including 21 tertiary referral hospitals, 5 secondary referral hospitals, and 2 municipal general hospitals. Inclusion criteria were as follows: age ≥20 years, diagnosis of cancer, cancer pain with a visual analog scale (VAS) score ≥1, and the ability to understand and sign the informed consent form. Patients with pain not associated with cancer were excluded at the discretion of the attending

physician. Patients' demographic and clinical characteristics (age, sex, cancer duration, cancer stage, and comorbidities related to NCP) and their treatment history (radiotherapy, chemotherapy, and surgery) were collected from medical records. Questionnaires and case report forms were collected, and study data were analyzed. The study (study ID: KCSG PC13-02) was approved by the central institutional review board (IRB) of the Korean Cancer Study Group (KCSG). After central approval, institutional approval was obtained from the institutions of the participating investigators. All procedures were, therefore, performed in accordance with the ethical standards of the institutional and/or national research committee and within the guidelines of the 1964 Declaration of Helsinki and its later amendments or comparable ethical standards.

## Pain assessment

All patients were evaluated by physicians, who determined the presence of cancer pain using the VAS and the presence of NCP using the Douleur Neuropathic (DN4) questionnaire during patients' regular visits. We defined a VAS score $\geq 1$ as cancer pain. The DN4 questionnaire includes seven pain descriptors, namely, burning painful cold, electric shocks, tingling, pins and needles, numbness, and itching, along with three additional symptoms, namely, hypoesthesia to touch, hypoesthesia to pinpricks, and allodynia to a paintbrush. The former seven descriptors were assessed via patient interviews, while the latter three were assessed via standardized clinical examinations. We defined a DN4 score $\geq 4$ as NCP.

## Statistical analysis

Patients were categorized into NCP (DN4 score $\geq 4$) and non-NCP (DN4 score $< 4$) groups. In the univariable analysis, patients' demographic and clinical characteristics were compared between the NCP and non-NCP groups using Student's t-tests for continuous variables and chi-square tests for categorical variables. To identify independent predictors of NCP, multivariable logistic regression analysis was used; the multivariable model included patients' demographic characteristics (age and gender) and variables with a p-value of $< 0.1$ in the univariable analysis.

Differences in symptoms reported by patients through the DN4 questionnaire were analyzed between the NCP and non-NCP groups using logistic regression models. In both univariable and age- and sex-adjusted multivariable analysis, the non-NCP group was the designated reference group. The distribution of clinical predictors in the NCP group was analyzed using chi-square tests.

All reported p-values were two-tailed, and a p-value of $< 0.05$ was considered statistically significant. All statistical analyses were performed using IBM SPSS Statistics 20.0 (IBM Corporation, NY, USA).

## Results

### Baseline characteristics of the patients and clinical predictors of NCP

A total of 2,003 patients participated in this study, and NCP was present in 722 (36.0%) of these patients. According to univariable analysis, using Student's t-tests for continuous variables and chi-square tests for categorical variables, the proportion of NCP was higher among patients aged $< 65$ years compared to those aged $\geq 65$ years (p$< 0.001$), in patients with cancer duration $\geq 6$ months compared to $< 6$ months (p = 0.001), and in patients with stage I–III cancer compared to those with stage IV cancer (p = 0.021). There were 494 patients with comorbidities (309 patients with diabetes, 44 patients with liver cirrhosis, 43 patients with traumatic

**Table 1. Comparison of the characteristics between the NCP and non-NCP groups (N = 2,003).**

| N (%) | Total N = 2,003 | NCP N = 722 (%) | Non-NCP N = 1,281 (%) | P-value |
|---|---|---|---|---|
| **Gender** | | | | 0.281 |
| Male | 1,089 | 381 (35.0) | 708 (65.0) | |
| Female | 914 | 341 (37.3) | 573 (62.7) | |
| **Age (years), mean±SD** | 60.9±11.3 | 59.5±10.7 | 61.7±11.5 | <0.001 |
| <65 | 1,213 | 489 (40.3) | 724 (59.7) | 0.001 |
| ≥65 | 790 | 233 (29.5) | 557 (70.5) | |
| **Cancer duration (months), mean±SD** | 26.3±37.1 | 30.1±42.1 | 24.1±33.7 | 0.001 |
| <6 | 545 | 143 (26.2) | 402(73.8) | 0.001 |
| ≥6 | 1,458 | 579 (39.7) | 879(60.3) | |
| **Cancer stage** | | | | 0.021 |
| Stage I–III | 399 | 163 (40.9) | 236 (59.1) | |
| Stage IV | 1,428 | 494 (34.6) | 951 (66.6) | |
| **Comorbidities** | | | | 0.133 |
| None | 1,509 | 530 (35.1) | 979 (64.9) | |
| Present | 494 | 192 (38.9) | 302 (61.1) | |
| **Radiotherapy** | | | | 0.310 |
| Never received | 1,401 | 515 (36.8) | 886 (63.2) | |
| Received | 602 | 207 (34.4) | 395 (65.6) | |
| **Chemotherapy** | | | | <0.001 |
| Never received | 250 | 54 (21.6) | 196 (78.4) | |
| Received[†] | 1,751 | 668 (38.1) | 1,083 (61.9) | |
| **Cancer surgery** | | | | 0.003 |
| Did not undergo | 1,204 | 403 (33.5) | 801 (66.5) | |
| Underwent | 799 | 319 (39.9) | 480 (60.1) | |
| **Pain VAS score, mean±SD** | 4.4±2.3 | 4.9±2.3 | 4.1±2.0 | <0.001 |
| <4 (mild pain) | 830 | 225 (27.1) | 605(72.9) | <0.001 |
| ≥4 (moderate-to-severe pain) | 1,173 | 497 (42.4) | 676 (57.6) | |

$\chi^2$ test, Student's t-tests for continuous variables and chi-square tests for categorical variables.

NCP: Neuropathic cancer pain; SD: Standard deviation; VAS: Visual analog scale.

injury, 41 with herpes zoster) and the proportion of patients with comorbidities was not significantly different between the two groups (p = 0.133). Regarding treatment history, the proportion of patients with NCP was higher among patients with a history of chemotherapy (p<0.001) and among patients with a history of cancer surgery (p = 0.003). Finally, the proportion of patients with NCP was higher among patients with moderate-to-severe pain (VAS score ≥4) compared to those with mild pain (VAS score <4; p<0.001; Table 1).

Multivariable logistic regression analysis revealed that age <65 years compared to age ≥65 years [odds ratio (OR), 1.57; 95% confidence interval (CI), 1.27–1.93], cancer duration of >6 months compared to that of <6 months (OR, 1.57; 95% CI, 1.23–2.01), stage IV cancer compared to stage I–III cancer (OR, 0.75; 95% CI, 0.59–0.96), a previous or current history of chemotherapy (OR, 1.74; 95% CI, 1.23–2.47), and moderate-to-severe pain (VAS score ≥4; OR, 2.05; 95% CI, 1.67–2.52) were significant clinical predictors of NCP (Table 2).

## Frequency of the descriptive symptoms and signs of NCP

Overall, the most common neuropathic symptom was tingling (N = 1,114, 55.6%), followed by electric shocks (N = 1,069, 53.4%) and pins and needles (N = 844, 42.1%). Among the groups

**Table 2. Clinical predictors of patients with NCP (N = 2,003).**

| Predictors (Reference) | Adjusted OR | 95% CI | P-value |
|---|---|---|---|
| Female (Male) | 1.036 | 0.849–1.265 | 0.729 |
| Age <65 years (≥65 years) | 1.567 | 1.270–1.934 | <0.001 |
| Cancer duration ≥6 months (<6 months) | 1.574 | 1.232–2.012 | <0.001 |
| Cancer stage IV (I-III) | 0.752 | 0.593–0.955 | 0.019 |
| Chemotherapy, done (Never done) | 1.740 | 1.225–2.472 | 0.002 |
| Surgery, done (Never done) | 1.127 | 0.917–1.386 | 0.256 |
| Pain VAS score, Moderate/Severe (Mild) | 2.054 | 1.671–2.524 | <0.001 |

Logistic regression analysis, variables with p-value<0.1 from the univariable analysis were included.

CI: Confidence interval; NCP: Neuropathic cancer pain; OR: Odds ratio; VAS: Visual analog scale.

categorized on the basis of the DN4 questionnaire score, commonly experienced symptoms were tingling (NCP; N = 638 vs non-NCP; N = 476), electric shocks (NCP; N = 632 vs non-NCP; N = 437), and pins and needles (NCP; N = 471 vs non-NCP; N = 373). An increased OR was noted for all symptoms and signs described in the DN4 questionnaire, indicating the increased incidence of these symptoms in NCP patients in both univariable and multivariable models. Prick hypoesthesia (OR, 36.95; 95% CI, 20.88–65.38) and touch hypoesthesia (OR, 20.56; 95% CI, 15.79–26.78) were associated with increased odds of being present in the NCP group by chi-square tests (Table 3).

## Clinical symptoms and signs of NCP according to each predictor

The prevalence of the clinical predictors (age <65 years, cancer duration ≥6 months, stage I–III cancer, history of chemotherapy, VAS score for pain ≥4) was assessed among the patients in the NCP group for each of the clinical symptoms and signs (Table 4). Among NCP patients with hyperesthesia, the proportion of patients aged <65 years (74.6%) was significantly higher than that of patients aged ≥65 years (25.4%) (p = 0.024). The proportion of patients with cancer duration of ≥6 months was significantly higher among NCP patients presenting with

**Table 3. Differences in selecting symptom items of the DN4 questionnaire between the NCP (N = 722) and non-NCP (N = 1,281) groups.**

| DN4 Symptom items | Total | NCP | Non-NCP | Univariable | Multivariable* | |
|---|---|---|---|---|---|---|
| | N = 2,003(%) | N = 722 (%) | N = 1,281 (%) | Crude OR (95% CI) | Adjusted OR (95% CI) | Pseudo R²** |
| Burning | 312 (15.6) | 198 (63.5) | 114 (36.5) | 3.87 (3.01–4.98) | 3.86 (3.00–4.97) | 0.088 |
| Painful Cold | 450 (22.5) | 345 (76.7) | 105 (23.3) | 10.25 (8.00–13.13) | 10.51 (8.18–13.48) | 0.265 |
| Electric shocks | 1,069 (53.4) | 632 (59.1) | 437 (40.9) | 13.56 (10.57–17.40) | 13.35 (10.40–17.14) | 0.348 |
| Tingling | 1,114 (55.6) | 638 (57.3) | 476 (42.7) | 12.85 (9.96–16.56) | 12.69 (9.84–16.37) | 0.330 |
| Pins and Needles | 844 (42.1) | 471 (55.8) | 373 (44.2) | 4.57 (3.76–5.55) | 4.53 (3.72–5.51) | 0.168 |
| Numbness | 512 (25.6) | 398 (77.7) | 114 (22.3) | 12.58 (9.87–16.02) | 12.57 (9.86–16.03) | 0.318 |
| Itching | 328 (16.4) | 220 (67.1) | 108 (32.9) | 4.76 (3.70–6.13) | 4.64 (3.60–5.98) | 0.111 |
| Touch hypoesthesia | 524 (26.2) | 436 (83.2) | 88 (16.8) | 20.67 (15.89–26.88) | 20.56 (15.79–26.78) | 0.405 |
| Prick hypoesthesia | 213 (10.6) | 200 (93.9) | 13 (6.1) | 37.37 (21.13–66.09) | 36.95 (20.88–65.38) | 0.233 |
| Hyperesthesia | 238 (11.9) | 189 (79.4) | 49 (20.6) | 8.92 (6.41–12.40) | 8.75 (6.29–12.19) | 0.147 |

Chi-square tests, all statistically significant, P<0.001.

*Age- and sex-adjusted

**Negelkerke R².

CI: Confidence interval; NCP: Neuropathic cancer pain; OR: Odds ratio.

**Table 4. Characteristics of neuropathic pain only in NCP patients in the presence or absence of the clinical predictors of NCP[†].**

| Total | Age (years) (N = 722) | | | Cancer duration (months) (N = 722) | | | Cancer stage (N = 657) | | | Chemotherapy (N = 722) | | | Pain VAS (N = 722) | | |
|---|---|---|---|---|---|---|---|---|---|---|---|---|---|---|---|
| | <65 (N = 489) | ≥65 (N = 233) | P-value | ≥6 (N = 579) | <6 (N = 143) | P-value | I–III (N = 163) | IV (N = 494) | P-value | Yes (N = 668) | No (N = 54) | P-value | ≥4 (N = 497) | <4 (N = 225) | P-value |
| Burning | 128 (64.6) | 70 (35.4) | 0.317 | 156 (78.8) | 42 (21.2) | 0.633 | 41 (22.3) | 143 (77.7) | 0.404 | 180 (90.9) | 18 (9.1) | 0.393 | 146 (73.7) | 52 (26.3) | 0.097 |
| Painful cold | 229 (66.4) | 116 (33.6) | 0.507 | 272 (78.8) | 73 (21.2) | 0.436 | 74 (23.7) | 238 (76.3) | 0.599 | 318 (92.2) | 27 (7.8) | 0.844 | 266 (77.1) | 79 (22.9) | <0.001 |
| Electric shocks | 420 (66.5) | 212 (33.5) | 0.069 | 510 (80.7) | 122 (19.3) | 0.450 | 143 (25.0) | 430 (75.0) | 0.927 | 587 (92.9) | 45 (7.1) | 0.449 | 439 (69.5) | 193 (30.5) | 0.401 |
| Tingling | 437 (68.5) | 201 (31.5) | 0.276 | 520 (81.5) | 118 (18.5) | 0.022 | 146 (25.0) | 438 (75.0) | 0.861 | 591 (92.6) | 47 (7.4) | 0.924 | 439 (68.8) | 199 (31.2) | 1.000 |
| Pins and needles | 314 (66.7) | 157 (33.3) | 0.452 | 373 (79.2) | 98 (20.8) | 0.409 | 108 (24.9) | 325 (75.1) | 0.989 | 432 (91.7) | 39 (8.3) | 0.331 | 345 (73.2) | 126 (26.8) | 0.001 |
| Numbness | 271 (68.1) | 127 (31.9) | 0.88 | 331 (83.2) | 67 (16.8) | 0.033 | 95 (26.2) | 268 (73.8) | 0.420 | 367 (92.2) | 31 (7.8) | 0.835 | 275 (69.1) | 123 (30.9) | 0.932 |
| Itching | 154 (70.0) | 66 (30.0) | 0.437 | 176 (80.0) | 44 (20.0) | 1.000 | 55 (26.6) | 152 (73.4) | 0.541 | 206 (93.6) | 14 (6.4) | 0.548 | 147 (66.8) | 73 (33.2) | 0.492 |
| Touch hypoesthesia | 299 (68.6) | 137 (31.4) | 0.602 | 362 (83.0) | 74 (17.0) | 0.024 | 102 (25.6) | 296 (74.4) | 0.610 | 409 (93.8) | 27 (6.2) | 0.139 | 291 (66.7) | 145 (33.3) | 0.156 |
| Prick hypoesthesia | 136 (68.0) | 64 (32.0) | 0.994 | 166 (83.0) | 34 (17.0) | 0.286 | 52 (28.0) | 134 (72.0) | 0.283 | 187 (93.5) | 13 (6.5) | 0.645 | 138 (69.0) | 62 (31.0) | 1.000 |
| Hyperesthesia | 141 (74.6) | 48 (25.4) | 0.024 | 155 (82.0) | 34 (18.0) | 0.533 | 37 (22.3) | 129 (77.7) | 0.444 | 173 (91.5) | 16 (8.5) | 0.661 | 135 (71.4) | 54 (28.6) | 0.421 |

$\chi^2$ test

[†] Multiple-choice response.

NCP: Neuropathic cancer pain; VAS: Visual analog scale.

tingling (81.5%), numbness (83.2%), or touch hypoesthesia (83.0%) (p = 0.022, 0.033, 0.024, respectively). NCP patients with painful cold and pins and needles were more likely to have moderate-to-severe pain (p<0.001, p = 0.001, respectively). There were no significant differences between patients with early-stage cancer (stage I–III) and those with later-stage cancer (stage IV) or between patients with a history of chemotherapy and those without a history of chemotherapy (Table 4).

## Discussion

In the present study, we analyzed data from a nationwide cancer pain survey in Korea, focusing on predictors of NCP and the clinical significance of patient-reported symptoms and signs most relevant for these predictors in Korean patients. Our study identified five clinical predictors of NCP: age <65 years, a past or current history of chemotherapy, cancer duration of ≥6 months, moderate-to-severe pain (VAS score ≥4), and early-stage cancer (stage I–III). Bulls et al. reported that older adults (aged ≥65 years) with gynecologic cancer are at higher risk for NCP [13]. Other studies have also reported a higher prevalence of neuropathy in older patients [14,15]. In the present study, 20% of the older (≥65 years) and 7% of the younger (<65 years) age groups had not received chemotherapy (S1 Table). Older patients are less likely to receive chemotherapy, which can explain the increased association of NCP with the younger age group in our study. A recent study reported that the predictors of NCP were associated with the location of neural invasion, a longer disease duration, a higher pain intensity, recent surgery or chemotherapy, the use of one or more adjuvant analgesics, and the presence of an episodic incident or breakthrough pain [16]. This was similar to our study, where NCP was found to be associated with chemotherapy, moderate-to-severe pain and longer duration of cancer.

However, in our study, surgery had a significant association with NCP in univariate, but not in multivariate, analysis. There were 399 patients with stage I, II, or III cancer and 799 patients underwent surgery. About half of the patients who underwent surgery had palliative surgery rather than curative surgery on the primary tumor, which may explain the difference from the previous report. Additionally, we enrolled various cancer patients in this study. Gastrointestinal cancer was the most common (49% of patients), followed by respiratory cancer (24%), skin, bone, and connective cancer (14%), hematologic malignancy (14%), genitourinary cancer (13%), breast cancer (11%), and head and neck cancer (9%) [10]. Since approximately half of the patients registered in this study had gastrointestinal cancer, it indicates that there are fewer patients who have undergone chronic pain-related surgeries, such as mastectomy or thoracotomy [17]. Moreover, less invasive surgical techniques might result in less postsurgical pain. Lumpectomy might result in more pain than a modified radical mastectomy [18].

Early neuropathic symptoms and signs of platinum analog-treated patients include loss of ankle jerks and decreased vibratory sensibility in the toes associated with numbness, tingling or paresthesias in finger and toes and prolonged treatment may worsen those symptoms. Pin and temperature sensation, joint position and light touch perception are less severely affected [19,20]. In our study, the most frequently reported symptoms were tingling, electric shock, and pins and needles, and these symptoms were most frequently reported by both NCP and non-NCP groups. It is possible for neuropathic pain to begin with common symptoms such as tingling, pins and needles and electric shocks. A Spanish prospective study, conducted over 1 month, that recruited 366 patients with cancer, aged ≥18 years, diagnosed with NCP (DN4 ≥4), and with moderate-to-severe pain (VAS ≥4), reported that the most common symptoms were tingling (79.8%), pins and needles (76.2%) and electric shocks (72.4%) [21]. This showed a similar distribution of descriptive symptoms as the NCP group in the present study. In contrast, an Indian study reported that a pricking type of pain was the most characteristic feature

(47.8%), followed by shooting pain (38.3%), in patients with NCP [22]. These differences in descriptive symptoms might be associated with differences in study design, ethnicity, or expressive language used to express patients' pain. The symptoms and signs of highest relevance to NCP were prick hypoesthesia and touch hypoesthesia. Further studies to clarify symptom definitions and to elucidate the pathogenesis of NCP are needed for its effective management.

The evaluation of pain begins with a patient interview regarding the symptoms. Currently, there is no gold standard method for the diagnosis of neuropathic pain. In daily practice, screening questionnaires are suitable for identifying potential patients with neuropathic pain. For cancer patients, these questionnaires are preferred over imaging studies such as electroneuromyography or magnetic resonance imaging. A screening tool for neuropathic pain could alert physicians about the possibility of NCP. Medical oncologists generally use three screening tools for neuropathic pain, namely pain DETECT [23], the Leeds Assessment of Neuropathic Symptoms and Signs (LANSS) [24], and the DN4 questionnaire [25]. Among these, only LANSS and the DN4 questionnaire feature items related to sensory examination. The LANSS was more specific, although less sensitive, than the DN4. The DN4 questionnaire is more sensitive than LANSS, regardless of the pain severity [26].

Chemotherapy-induced peripheral neuropathy (CIPN) is a frequent, dose-dependent complication of anticancer drugs, including platinum, taxanes, epothilones, vinca alkaloids, and newer agents such as bortezomib [27]. In our study, of the 722 patients with NCP, 93% had a history of chemotherapy. Although anticancer chemotherapeutics can intensify painful peripheral neuropathy, there was no significant difference in the symptomatology of NCP between patients with and without a history of chemotherapy. However, some symptoms and signs were related to specific predictors. Hyperesthesia was more frequent in patients aged <65 years than in those aged ≥65 years. Tingling and touch hypoesthesia were more frequent in patients with longer cancer duration (≥6 months). And painful cold and pins and needles were associated with moderate and severe cancer pain. To our knowledge, this study is very meaningful as it analyzed each of the symptoms and signs based on the predictors of NCP. These finding could help physicians in the early diagnosis and management of NCP.

This study has some limitations. First, the cross-sectional design limits our knowledge of various pre-referral treatments and their degree of consistency with standard guidelines. Second, the origin of cancer, stage of cancer, and anticancer treatment differed among the enrolled patients. There is controversy over the association of status or site of cancer with NCP [14,28], rather, whether NCP is associated with neurological involvement secondary to the disease or its treatment. Third, we did not differentiate dominant or neuropathic components from mixed type with nociceptive and neuropathic pain. As we searched for the frequency and risk factors of NCP in all patients with cancer pain through cross-sectional studies, we were not able to study the cause of NCP. This is required through other studies in the future. Fourth, the patients' narratives, including their interpretation of pain, may be governed by their culture and language [29]. Finally, we defined that patients had cancer pain if they complained of cancer pain (NRS ≥1), regardless of treatment. Patients with pain more than VAS 3 were usually considered for treatment in clinical practice. There might be limits to the application of treatment in clinical practice. However, this study is helpful in identifying patients who are expected to have NCP in advance and for monitoring their symptoms and signs by risk factors.

In conclusion, the results of this nationwide cross-sectional study in Korea have shown predictors of NCP and the identification of symptoms and signs associated with NCP. To the best of our knowledge, this is the first study that has analyzed each descriptive symptom and sign based on the predictors of NCP, as obtained by the DN4 questionnaire. More studies to

substantiate the characteristics of NCP could help in its early detection, correct diagnosis and efficient management.

## Supporting information

**S1 Table. Comparison of patients receiving/not receiving chemotherapy and age <65 or ≥65 years.**
(DOCX)

## Acknowledgments

On behalf of the Korean Cancer Study Group (KCSG), we thank the investigators who participated in this study: So Yeon Oh, Dae Sik Hong, Kyung Hee Lee, So Young Yoon, Jung Lim Lee, Hyun Chang, Keon Uk Park, Young Seon Hong, Sung Rok Kim, Na Ri Lee, Jeanno Park, Sang Cheul Oh, Seung-Sei Lee, Dae Ro Choi, Jooseop Chung, In Sil Choi, and Joung Soon Jang. In addition, we appreciate the assistance in statistical analysis provided by Sook Jung Hyun, Professor of Health Administration, Department of Management & Administration, Baekseok Arts University. Editorial assistance was provided by David P. Figgitt PhD, ISMPP CMPP™, Content Ed Net, with funding from Viatris.

## Author Contributions

**Conceptualization:** Sang Won Shin.

**Data curation:** Sun Kyung Baek, Sang Won Shin, Su-Jin Koh, Jung Han Kim, Hyo Jung Kim, Byoung Yong Shim, Seok Yun Kang, Sang Byung Bae, Hwan Jung Yun, Sun Jin Sym, Hye Sook Han.

**Formal analysis:** Sang Won Shin, Ha Yeong Gil.

**Investigation:** Sun Kyung Baek, Sang Won Shin, Su-Jin Koh, Jung Han Kim, Hyo Jung Kim, Byoung Yong Shim, Seok Yun Kang, Sang Byung Bae, Hwan Jung Yun, Sun Jin Sym, Hye Sook Han.

**Methodology:** Su-Jin Koh, Jung Han Kim, Hyo Jung Kim, Byoung Yong Shim, Seok Yun Kang, Sang Byung Bae, Hwan Jung Yun, Sun Jin Sym, Hye Sook Han.

**Resources:** Sang Won Shin, Su-Jin Koh, Jung Han Kim, Hyo Jung Kim, Byoung Yong Shim, Seok Yun Kang, Sang Byung Bae, Hwan Jung Yun, Sun Jin Sym, Hye Sook Han.

**Supervision:** Sang Won Shin, Su-Jin Koh, Jung Han Kim, Hyo Jung Kim, Byoung Yong Shim, Seok Yun Kang, Sang Byung Bae, Hwan Jung Yun, Sun Jin Sym, Hye Sook Han.

**Visualization:** Su-Jin Koh, Jung Han Kim, Hyo Jung Kim, Byoung Yong Shim, Seok Yun Kang, Sang Byung Bae, Hwan Jung Yun, Sun Jin Sym, Hye Sook Han.

**Writing – original draft:** Sun Kyung Baek, Sang Won Shin.

**Writing – review & editing:** Sang Won Shin, Su-Jin Koh, Jung Han Kim, Hyo Jung Kim, Byoung Yong Shim, Seok Yun Kang, Sang Byung Bae, Hwan Jung Yun, Sun Jin Sym, Hye Sook Han, Ha Yeong Gil.

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
