## [Decision Letter · Decision Letter 0]

5 Feb 2021

PONE-D-20-33583

Significance of descriptive symptoms and signs and clinical parameters as predictors of neuropathic cancer pain

PLOS ONE

Dear Dr. Shin,

Thank you for submitting your manuscript to PLOS ONE. After careful consideration, we feel that it has merit but does not fully meet PLOS ONE’s publication criteria as it currently stands. Therefore, we invite you to submit a revised version of the manuscript that addresses the points raised during the review process.

We look forward to receiving your revised manuscript.

Kind regards,

Ahmed Negida, MD

Academic Editor

PLOS ONE

Journal Requirements:

'This research was sponsored by Pfizer.'

We note that one or more of the authors have an affiliation to the commercial funders of this research study: Pfizer

Reviewers' comments:

Reviewer's Responses to Questions

**Comments to the Author**

1. Is the manuscript technically sound, and do the data support the conclusions?

Reviewer #1: Partly

Reviewer #2: Yes

Reviewer #3: Partly

Reviewer #4: Partly

2. Has the statistical analysis been performed appropriately and rigorously? 

Reviewer #1: Yes

Reviewer #2: Yes

Reviewer #3: No

Reviewer #4: Yes

3. Have the authors made all data underlying the findings in their manuscript fully available?

Reviewer #1: No

Reviewer #2: Yes

Reviewer #3: Yes

Reviewer #4: Yes

4. Is the manuscript presented in an intelligible fashion and written in standard English?

Reviewer #1: Yes

Reviewer #2: Yes

Reviewer #3: No

Reviewer #4: Yes

5. Review Comments to the Author

Reviewer #1: The manuscript by Baek et al. aims to identify new predictors of NCP in Korean population of cancer patients by using the DN4. Although authors identified some signs and symptoms associated with NCP, the main objective and output are not enough empathised leaving in the reader a feeling of non-understanding the relevance of the results.

Overall, this is an interesting attempt, but there are some points to address:

1) Line 72: authors mentioned that the prevalence of NCP is 20-30% in cancer patients, worldwide? In Korea? Please specify.

2) Line 81: according to authors, few studies have reported how patients described the influence of their symptoms, however, no reference is provided. I would appreciate to see the references.

3) Patients aged >20 years are included in the present study. However, results only show the comparison between <65 and > 65. Could authors explain the reason for this selection? Have the authors performed any previous analysis to define this age categories?

4) In order to facilitate the comprehensive interpretation of the results, I would appreciate the authors to mention which statistical method has been used for each analysis in the results section, including figures and tables, as it is only mentioned in “Statistical analysis”.

5) Concerning the Figure 1: please consider that P values less than 0.001 are summarized with three asterisks (***) not two (**).

There is not Title nor footnotes in Figure 1.

6) Please add a reference in line 203 that confirms that the presence of comorbidities is higher in the elderly.

7) Reference 9 in line 206 does not correspond to the sentence as it mentions the results of the current manuscript. Please check.

8) In the discussion, authors explain that the longer duration and higher intensity of the treatment received in the younger patients could explain the results. However, there is no information about the treatment dose or the treatment duration in the sample analysed. Therefore, I would recommend showing this information in a table or to elaborate another explanation for the result found.

9) If, as mentioned in the discussion, DN4 may have limitations when used in patients with severe pain, why authors have decided to use this questionnaire to evaluate their participants if more than 50% presented moderate to severe pain? Are they sure that this is the appropriate tool for this study? Please justify.

10) Has been data provided or deposited to a public repository? If not, more detailed information should be added to the statistics (behind means, medians…).

Reviewer #2: Manuscript Number: PONE-D-20-33583

Title: Significance of descriptive symptoms and signs and clinical parameters as predictors

of neuropathic cancer pain

Summary

I appreciate the opportunity to review this interesting report. This is a nationwide cross-sectional study in Korea that have shown predictors of NCP and the identification of symptoms and signs associated with NCP.

Major Strengths

This is an interesting article that first describe symptoms and signs based on the predictors of NCP, as obtained by the DN4 questionnaire using nationwide data.

Major Weakness

Please describe more specific sources and methods of selection of participants. (Ex, hospital type, department, visit number…)

The authors indicate that the inclusion criteria were cancer pain with VAS ≥ 1. I think the VAS 1 is not suitable. Is there any reason?

In my opinion, the diagnosis of NCP is more complex one. Especially in the cancer pain population, in which nociceptive and neuropathic components frequently co-exist. How did the authors differentiate NCP with mixed type pain?

In the discussion, page 18, line 266-267, is there any reference? As I know, NCP may occur directly as in tumor-associated neural compression.

According to the previous report, surgery is one of predictor of NCP. Please discuss more about the predictor of NCP.

Minor Comments

According to the STROBE guideline, the title represents “Indicate the study’s design”.

Reviewer #3: Authors have presented an analysis of signs and symptoms of cancer pain in a large sample size. The topic is surely interesting and useful for clinicians in order to improve diagnosis of cancer pain. Authors (as they stated in the background) have already published part of the data in a previous paper and some data seem the same. Please control accurately it.

The major concern regarding the manuscript is the selection of the patients: it seems that a VAS>1 could lead to wrong results (even though we have no any informations about drugs that patients took).

I would suggest to consider only patients with pain more than 3 (as usually considered in literature), eventually stratifying the population in mild (VAS 4-6) and severe (VAS 7-10) pain. This stratification could be more reliable to clinical practice and could be more helpful for a clear analysis of the samples size.

I have also some concerns about the statistical analysis and the data presentation. It should be better to evaluate if there are differences in percentages of patients among the same group (NCP and no NCP). As presented, it seems that authors have evaluated differences between NCP and no NCP for the single variable. This analysis could be misunderstanding also considering also the huge heterogeneity of the sample size. Please discuss it

Can we have some data about use of analgesic drugs and eventually differences in type of drugs used and pain relief obtained in the two groups?it could be helpful to understand if a better analysis of the signs and symptoms could improve the effectiveness of treatment.

Finally it should be necessary to revise English and revise several typos.

Reviewer #4: Thanks for opportunity of reviewing this article. This is an interesting study, but there are several limitations with NCP predictor.

Method

1.In method section, explanation of study design and definition is lacking.

2.What are the criteria for determining NCP and Non-NCP? The criteria for diagnosis are unclear.

3.P6 L 105 “A VAS score ≥1 was indicative of cancer pain.” Do you have a reference? Is this the standard set by the authors? Are they defined by the authors?

Result

Table 1

1.Comorbidity is also an important factor in determining cancer pain. For example, the case of patient with severe DM, the sensitivity of pain decreases and pain patterns may be different due to DM neuropathy. Could you add details about Comorbidity?

2.The type of cancer is also an important factor. If you can, add it.

3.The type of cancer is also an important factor. If you can, add it.

Cancer patients often use narcotic analgesics, which can also lead to hyperalgesia if prolonged exposure to analgesics. Add the presence or absence of narcotic pain killers.

4.P8 L139, Table 1

The standard for dividing the VAS score is set to 4, do you have any reference?

Discussion

In the discussion section, the discussion is too long, and the content of the sentence is in conflict. Need to clean up.

1. What is the clinical meaning of this investigation result?

2. P15 199-206, I cannot understand your explanation

3. According to your results, chemotherapy is also a risk factor for NCP. Old age receives less chemotherapy than young age, so why have more NCP in old age? Does cancer pain decrease in young age receiving toxic agent chemotherapy?This section conflicts with your results and needs clarification. In addition, check the reference again.

4. P16 L217-239 Combine in one paragraph.

Delete "Failure of identifying NCP leads to unsatisfactory pain management in Asian countries where physician–patient communication tends to be reserved or of short duration"(??)

6. PLOS authors have the option to publish the peer review history of their article (what does this mean?). If published, this will include your full peer review and any attached files.

Reviewer #1: No

Reviewer #2: No

Reviewer #3: No

Reviewer #4: No

---

## [Author Response · Author response to Decision Letter 0]

13 Apr 2021

29 March, 2021

Dear Editor-in-Chief, PLOS ONE

We deeply appreciate your insightful comments on our manuscript entitled, “Significance of descriptive symptoms and signs and clinical parameters as predictors of neuropathic cancer pain”. Your comments were really helpful for revising our manuscript. We totally agree with the comments of the reviewers and have revised our manuscript according to reviewers’ comments. All relevant data are within the manuscript, and the changes and responses to specific comments are shown below. We hope that these changes will satisfy the editors and reviewers. 

Also, we confirm that this study was sponsored by Pfizer Pharmaceuticals Korea Ltd. Ha Yeong Gil, the employee of Pfizer Pharmaceuticals Korea Ltd., contributed in the study design, data analysis, decision to publish, or preparation of the manuscript. We declare that the authors have no other competing interests. This does not alter our adherence to all the PLOS ONE policies on sharing data and materials. 

Thank you again for providing important comments on our work. We hope you can consider this paper for publication in PLOS ONE.

Sincerely,

Sang Won Shin, MD, PhD.

Medical Oncology, Department of Internal Medicine 

Korea University Anam Hospital, Korea University College of Medicine, Seoul, Korea

73, Inchon-ro, Seongbuk-gu, Seoul, 136-705, Republic of Korea 

Tel:+82-2-920-5350, Fax: +82-2-920-6951

E-mail: shinsw@kumc.or.kr

Response to the Referee

29 March, 2021

Thank you very much for your comments regarding our manuscript. We answered your comments with point-by-point descriptions and changed the manuscript accordingly.

Reviewer #1: The manuscript by Baek et al. aims to identify new predictors of NCP in Korean population of cancer patients by using the DN4. Although authors identified some signs and symptoms associated with NCP, the main objective and output are not enough empathised leaving in the reader a feeling of non-understanding the relevance of the results. 

Response: We would like to thank you for the careful comments. Based on the reviewer’s suggestion, we revised our paper, again. The revised words and sentences are red colored.

Overall, this is an interesting attempt, but there are some points to address:

1) Line 72: authors mentioned that the prevalence of NCP is 20-30% in cancer patients, worldwide? In Korea? Please specify.

Response: We would like to thank you for the careful comments. As you advised, the references have been reinforced and the sentence was modified as follows. 

“Neuropathic pain affects up to 40% of cancer patients and is associated with increased pain intensity and decreased quality of life (QOL) [7-9]. Previously, we also reported the incidence rate of NCP in cancer patients in Korea and the relevance that NCP deteriorates patients’ QOL [10].” (line 77-82)

Added references: 

[8] Bennett MI, Rayment C, Hjermstad M, et al. (2012) Prevalence and aetiology of neuropathic pain in cancer patients: a systematic review. Pain 153: 359-365.

[9] Rayment C, Hjermstad MJ, Aass N, et al. (2013) Neuropathic cancer pain: prevalence, severity, analgesics and impact from the European Palliative Care Research Collaborative-Computerised Symptom Assessment study. Palliat Med 27: 714-721.

2) Line 81: according to authors, few studies have reported how patients described the influence of their symptoms, however, no reference is provided. I would appreciate to see the references.

Response: We would like to thank you for the careful comments. We have excluded the sentence because it seems to cause confusion.

3) Patients aged >20 years are included in the present study. However, results only show the comparison between <65 and > 65. Could authors explain the reason for this selection? Have the authors performed any previous analysis to define this age categories?

Response: Thank you for the helpful comments. More than half of new cancer cases are diagnosed in adults over the age of 65 years. Older cancer patients are at high risk for chemotherapy-related toxicities including NCP. Bulls et al evaluated the longitudinal change in patients with NCP in older (>65) and younger (<65) patients. So, we compared patients based on age 65 years. We have added the article of Bulls et al as a reference. 

“Our study identified five clinical predictors of NCP: age <65 years, a past or current history of chemotherapy, cancer duration of ≥6 months, moderate-to-severe pain (VAS score ≥4), and early-stage cancer (stage I–III). Bulls et al. reported that older adults (aged ≥65 year) with gynecologic cancer are at higher risk for NCP [13].” (line 217-220)

Added reference: 

[13] Bulls HW, Hoogland AI, Kennedy B, et al. (2019) A longitudinal examination of associations between age and chemotherapy-induced peripheral neuropathy in patients with gynecologic cancer. Gynecol Oncol 152: 310-315.

4) In order to facilitate the comprehensive interpretation of the results, I would appreciate the authors to mention which statistical method has been used for each analysis in the results section, including figures and tables, as it is only mentioned in “Statistical analysis”.

Response: We would like to thank you for the careful comments. As you advised, we have added a sentence about statistical analysis in the results section, including figures and tables.

“According to univariable analysis, using Student’s t-tests for continuous variables and chi-square tests for categorical variables, the proportion of NCP was higher among patients aged <65 years compared to those aged ≥65 years (p<0.001), in patients with cancer duration ≥6 months compared to <6 months (p=0.001), and in patients with stage I–III cancer compared to those with stage IV cancer (p=0.021).” (line 143-147)

“Prick hypoesthesia (OR, 36.95; 95% CI, 20.88–65.38) and touch hypoesthesia (OR, 20.56; 95% CI, 15.79–26.78) were associated with increased odds of being present in the NCP group by chi-square tests (Table 3).” (line 186-188)

Table 1: “χ2 test, Student’s t-tests for continuous variables and chi-square tests for categorical variables” (line 159-160) 

Table 2: “Logistic regression analysis, variables with p-value<0.1 from the univariable analysis were included.” (line 172-173)

Table 3: “Chi-square tests, all statistically significant” (line 190)

5) Concerning the Figure 1: please consider that P values less than 0.001 are summarized with three asterisks (***) not two (**). There is not Title nor footnotes in Figure 1.

Response: Thank you for your helpful comments. We have converted Figure 1 to Table 2 to explain the Results more easily.

Table 2. Clinical predictors of patients with NCP (N=2,003)

Predictors(Reference) Adjusted OR 95% - CI P-value

Female (Male) 1.036 0.849 - 1.265 0.729

Age <65 years (≥65 years) 1.567 1.270 - 1.934 <0.001

Cancer duration≥6 months (<6 months) 1.574 1.232 - 2.012 <0.001

Cancer stage IV (I-III) 0.752 0.593 - 0.955 0.019

Chemotherapy, done (Never done) 1.740 1.225 - 2.472 0.002

Surgery, done (Never done) 1.127 0.917 - 1.386 0.256

Pain VAS score, Moderate/Severe (Mild) 2.054 1.671 - 2.524 <0.001

Logistic regression analysis, variables with p-value<0.1 from the univariable analysis were included. 

CI: confidence interval; NCP: neuropathic cancer pain; OR: odds ratio; VAS: visual analog scale

6) Please add a reference in line 203 that confirms that the presence of comorbidities is higher in the elderly.

Response: We would like to thank you for the careful comments. We excluded the sentence while revising the Discussion section. 

7) Reference 9 in line 206 does not correspond to the sentence as it mentions the results of the current manuscript. Please check.

Response: We would like to thank you for the careful comments. We removed the reference which did not correspond to the sentence.

8) In the discussion, authors explain that the longer duration and higher intensity of the treatment received in the younger patients could explain the results. However, there is no information about the treatment dose or the treatment duration in the sample analysed. Therefore, I would recommend showing this information in a table or to elaborate another explanation for the result found.

Response: We would like to thank you for the careful comments. We added Supplementary Table1 and revised the sentence.

“In the present study, 20% of the older (≥65 years) and 7% of the younger (<65 years) age groups had not received chemotherapy (S1 Table). Older patients are less likely to receive chemotherapy, which can explain the increased association of NCP with the younger age group in our study.” (line 228-232)

S1 Table. Comparison of patients receiving/not receiving chemotherapy and age <65 or ≥65 years

 Chemotherapy Non-chemotherapy Total P-value

Age <65 years 1,123 (92.6) 90 (7.4) 1,213 (60.6) <0.001

Age ≥65 years 630 (79.7) 160 (20.3) 790 (39.4) 

9) If, as mentioned in the discussion, DN4 may have limitations when used in patients with severe pain, why authors have decided to use this questionnaire to evaluate their participants if more than 50% presented moderate to severe pain? Are they sure that this is the appropriate tool for this study? Please justify.

Response: We would like to thank you for the careful comments. Unfortunately, the quote seems to have been wrong. Perez C et al reported the LANSS is considerably more specific than the DN4 if pain is severe and intermittent, not moderate and severe. We removed the sentence.

10) Has been data provided or deposited to a public repository? If not, more detailed information should be added to the statistics (behind means, medians…).

Response: We would like to thank you for the careful comments. We have added information about mean and standard deviations in Table 1. All relevant data are within the paper.

 

Table 1. Comparison of the characteristics between the NCP and non-NCP groups (N=2,003)

N(%) Total

N=2,003 NCP

N=722 (%) Non-NCP

N=1,281 (%) P-value

Gender 0.281

Male 1,089 381 (35.0) 708 (65.0) 

Female 914 341 (37.3) 573 (62.7) 

Age (years), mean�SD 60.9±11.3 59.5±10.7 61.7±11.5 <0.001

<65 1,213 489 (40.3) 724 (59.7) 0.001

≥65 790 233 (29.5) 557 (70.5) 

Cancer duration (months), mean�SD 26.3±37.1 30.1±42.1 24.1±33.7 0.001

<6 545 143 (26.2) 402(73.8) 0.001

≥6 1,458 579 (39.7) 879(60.3) 

Cancer stage 0.021

Stage I–III 399 163 (40.9) 236 (59.1) 

Stage IV 1,428 494 (34.6) 951 (66.6) 

Comorbidities 0.133

None 1,509 530 (35.1) 979 (64.9) 

Present 494 192 (38.9) 302 (61.1) 

Radiotherapy 0.310

Never received 1,401 515 (36.8) 886 (63.2) 

Received 602 207 (34.4) 395 (65.6) 

Chemotherapy <0.001

Never received 250 54 (21.6) 196 (78.4) 

Received† 1,751 668 (38.1) 1,083 (61.9) 

Cancer surgery 0.003

Did not undergo 1,204 403 (33.5) 801 (66.5) 

Underwent 799 319 (39.9) 480 (60.1) 

Pain VAS score, mean�SD 4.4±2.3 4.9±2.3 4.1±2.0 <0.001

<4 (mild pain) 830 225 (27.1) 605(72.9) <0.001

≥4 (moderate-to-severe pain) 1,173 497 (42.4) 676 (57.6) 

χ2 test, Student’s t-tests for continuous variables and chi-square tests for categorical variables 

NCP: neuropathic cancer pain; SD: standard deviation; VAS: visual analog scale

 

Reviewer #2: Manuscript Number: PONE-D-20-33583

Title: Significance of descriptive symptoms and signs and clinical parameters as predictors of neuropathic cancer pain

Summary

I appreciate the opportunity to review this interesting report. This is a nationwide cross-sectional study in Korea that have shown predictors of NCP and the identification of symptoms and signs associated with NCP.

Major Strengths

This is an interesting article that first describe symptoms and signs based on the predictors of NCP, as obtained by the DN4 questionnaire using nationwide data.

Major Weakness

Please describe more specific sources and methods of selection of participants. (Ex, hospital type, department, visit number…)

Response: We would like to thank you for the careful comments. We have added the requested information about hospitals.

“This nationwide, cross-sectional, descriptive, non-interventional, multicenter study was conducted from February 2013 to March 2014 in 28 hospitals in Korea, including 21 tertiary referral hospitals, 5 secondary referral hospitals, and 2 municipal general hospitals.” (line 96-99)

The authors indicate that the inclusion criteria were cancer pain with VAS ≥ 1. I think the VAS 1 is not suitable. Is there any reason? 

In my opinion, the diagnosis of NCP is more complex one. Especially in the cancer pain population, in which nociceptive and neuropathic components frequently co-exist. How did the authors differentiate NCP with mixed type pain?

Response: We totally agree that the diagnosis of NCP is more complex. We thought your opinion was very important. However, we were sorry that we did not differentiate NCP with mixed type pain or explore the cause of NCP. We have added this concern as a limitation.

“Third, we did not differentiate dominant or neuropathic components from mixed type with nociceptive and neuropathic pain. As we searched for the frequency and risk factors of NCP in all patients with cancer pain through cross-sectional studies and used DN4 as screening test of NCP, we were not able to study the cause of NCP. This is required through other studies in the future.” (line 310-313)

In the discussion, page 18, line 266-267, is there any reference? As I know, NCP may occur directly as in tumor-associated neural compression.

Response: We would like to thank you for the careful comments. We agree with your opinion, have revised the sentence, and added the reference.

“There is controversy over the association of status or site of cancer with NCP [14, 28], rather, whether NCP is associated with neurological involvement secondary to the disease or its treatment.” (line 307-309)

Added references:

[14] Lewis MA, Zhao F, Jones D, et al. Neuropathic Symptoms and Their Risk Factors in Medical Oncology Outpatients With Colorectal vs. Breast, Lung, or Prostate Cancer: Results From a Prospective Multicenter Study. J Pain Symptom Manage. 2015;49(6):1016-24. 

[28] Oosterling A, te Boveldt N, Verhagen C, et al. (2016) Neuropathic Pain Components in Patients with Cancer: Prevalence, Treatment, and Interference with Daily Activities. Pain Pract 16: 413-421.

According to the previous report, surgery is one of predictor of NCP. Please discuss more about the predictor of NCP.

Response: We would like to thank you for the careful comments. We have added one paragraph about your comments. 

“However, in our study, surgery had a significant association with NCP in univariate, but not in multivariate, analysis. There were 399 patients with stage I, II, or III cancer and 799 patients underwent surgery. About half of the patients who underwent surgery had palliative surgery rather than curative surgery on the primary tumor, which may explain the difference from the previous report. Additionally, we enrolled various cancer patients in this study. Gastrointestinal cancer was the most common (49% of patients), followed by respiratory cancer (24%), skin, bone, and connective cancer (14%), hematologic malignancy (14%), genitourinary cancer (13%), breast cancer (11%), and head and neck cancer (9%) [10]. Since approximately half of the patients registered in this study had gastrointestinal cancer, it indicates that there are fewer patients who have undergone chronic pain-related surgeries, such as mastectomy or thoracotomy [17]. Moreover, less invasive surgical techniques might result in less postsurgical pain. Lumpectomy might result in more pain than a modified radical mastectomy [18].” (line 241-254)

Minor Comments

According to the STROBE guideline, the title represents “Indicate the study’s design”.

Reviewer #3: Authors have presented an analysis of signs and symptoms of cancer pain in a large sample size. The topic is surely interesting and useful for clinicians in order to improve diagnosis of cancer pain. Authors (as they stated in the background) have already published part of the data in a previous paper and some data seem the same. Please control accurately it.

The major concern regarding the manuscript is the selection of the patients: it seems that a VAS>1 could lead to wrong results (even though we have no any informations about drugs that patients took).

I would suggest to consider only patients with pain more than 3 (as usually considered in literature), eventually stratifying the population in mild (VAS 4-6) and severe (VAS 7-10) pain. This stratification could be more reliable to clinical practice and could be more helpful for a clear analysis of the samples size.

Response: We totally agree with you. However, our research is a cross-sectional study that shows the frequency and risk factors of NCP in patients with cancer pain, and is limited by enrolling patients with VAS 1 or higher, regardless of treatment. We have included this in the limitations section. 

“Third, we did not differentiate dominant or neuropathic components from mixed type with nociceptive and neuropathic pain. As we searched for the frequency and risk factors of NCP in all patients with cancer pain through cross-sectional studies, we were not able to study the cause of NCP. This is required through other studies in the future. Fourth, the patients’ narratives, including their interpretation of pain, may be governed by their culture and language [29]. Finally, we defined that patients had cancer pain if they complained of cancer pain (NRS≥1) regardless of treatment. Patients with pain more than VAS 3 were usually considered for treatment in clinical practice. There might be limits to the application of treatment in clinical practice. However, this study is helpful in identifying patients who are expected to have NCP in advance and for monitoring their symptoms and signs by risk factors.” (line 310-325)

I have also some concerns about the statistical analysis and the data presentation. It should be better to evaluate if there are differences in percentages of patients among the same group (NCP and no NCP). As presented, it seems that authors have evaluated differences between NCP and no NCP for the single variable. This analysis could be misunderstanding also considering also the huge heterogeneity of the sample size. Please discuss it.

Response: We would like to thank you for the careful comments. However, we were sorry that we analyzed a difference between the risk factors of NCP and the descriptive symptoms (with or without NCP) in patients with cancer pain, which makes it difficult to see differences in the same group.

Can we have some data about use of analgesic drugs and eventually differences in type of drugs used and pain relief obtained in the two groups? it could be helpful to understand if a better analysis of the signs and symptoms could improve the effectiveness of treatment.

Response: We would like to thank you for the careful comments. Previously, we published a study on the association of NCP and QOL in the same patients, and a part relating to treatment was included in that paper (Oh SY, Shin SW, Koh SJ, Bae SB, Chang H, et al. Multicenter, cross-sectional observational study of the impact of neuropathic pain on quality of life in cancer patients. Support Care Cancer. 2017;25:3759-67). In this study, we presented the results that were not included in the previous study. The contents of treatment were reported in previous study. It is as follows (blue color) for your reference.

Previous article:

More than half of the patients (n= 1,173, 58.6%) had had moderate to severe pain (VAS≥4), implying that cancer pain was not adequately managed. Pharmacologic management for pain was as follows: 1,313 patients received opioids (65.6%), 748 received non-opioid analgesics (37.3%), 464 received anticonvulsants (23.2%), 134 received antidepressants (6.7%), 91 received corticosteroids (4.5%), 68 received benzodiazepines (3.4%), and 16 received phenothiazines (0.8%) (permitted overlap).

Finally it should be necessary to revise English and revise several typos.

Response: We would like to thank you for the careful comments. We have revised the English throughout the manuscript.

Reviewer #4: Thanks for opportunity of reviewing this article. This is an interesting study, but there are several limitations with NCP predictor.

Method

1. In method section, explanation of study design and definition is lacking.

2. What are the criteria for determining NCP and Non-NCP? The criteria for diagnosis are unclear.

3. P6 L 105 “A VAS score ≥1 was indicative of cancer pain.” Do you have a reference? Is this the standard set by the authors? Are they defined by the authors?

Response: We would like to thank you for the careful comments. We added the definition of cancer pain and NCP.

“We defined a VAS score ≥1 as cancer pain.” (line 115)

“We defined a DN4 score ≥4 as NCP.” (line 120-121)

Result

Table 1

1.Comorbidity is also an important factor in determining cancer pain. For example, the case of patient with severe DM, the sensitivity of pain decreases and pain patterns may be different due to DM neuropathy. Could you add details about Comorbidity?

2. The type of cancer is also an important factor. If you can, add it.

3. The type of cancer is also an important factor. If you can, add it.

Cancer patients often use narcotic analgesics, which can also lead to hyperalgesia if prolonged exposure to analgesics. Add the presence or absence of narcotic pain killers.

Response: We would like to thank you for the careful comments. Previously, we published a study on the association of NCP and QOL in the same patients, and a part relating to treatment was included in that paper (Oh SY, Shin SW, Koh SJ, Bae SB, Chang H, et al. Multicenter, cross-sectional observational study of the impact of neuropathic pain on quality of life in cancer patients. Support Care Cancer. 2017;25:3759-67). In this study, we presented results that were not included in previous study. Comorbidity and treatment were included in the previous report. We have briefly included comorbidity in the Results section and the type of cancer in the Discussion section. However, we were sorry that we could not include narcotic analgesics. For your reference, the contents of the previous paper on treatment are described below (blue color) 

Results section:

“There were 494 patients with comorbidities (309 patients with diabetes, 44 patients with liver cirrhosis, 43 patients with traumatic injury, 41 with herpes zoster) and the proportion of patients with comorbidities was not significantly different between the two group (p=0.133).” (line 147-150)

Discussion section:

“However, in our study, surgery had a significant association with NCP in univariate, but not in multivariate, analysis. There were 399 patients with stage I, II, or III cancer and 799 patients underwent surgery. About half of the patients who underwent surgery had palliative surgery rather than curative surgery on the primary tumor, which may explain the difference from the previous report. Additionally, we enrolled various cancer patients in this study. Gastrointestinal cancer was the most common (49% of patients), followed by respiratory cancer (24%), skin, bone, and connective cancer (14%), hematologic malignancy (14%), genitourinary cancer (13%), breast cancer (11%), and head and neck cancer (9%) [10]. Since approximately half of the patients registered in this study had gastrointestinal cancer, it indicates that there are fewer patients who have undergone chronic pain-related surgeries, such as mastectomy or thoracotomy [17]. Moreover, less invasive surgical techniques might result in less postsurgical pain. Lumpectomy might result in more pain than a modified radical mastectomy [18].” (line 241-254)

Previous article:

More than half of the patients (n= 1,173, 58.6%) had had moderate to severe pain (VAS≥4), implying that cancer pain was not adequately managed. Pharmacologic management for pain was as follows: 1,313 patients received opioids (65.6%), 748 received non-opioid analgesics (37.3%), 464 received anticonvulsants (23.2%), 134 received antidepressants (6.7%), 91 received corticosteroids (4.5%), 68 received benzodiazepines (3.4%), and 16 received phenothiazines (0.8%) (permitted overlap).

4. P8 L139, Table 1

The standard for dividing the VAS score is set to 4, do you have any reference?

Response: We would like to thank you for the careful comments. We divided patients based on VAS 4 because patients with VAS 4 or more need treatment. There is likely a controversy. We have added that as a limitation. 

“Finally, we defined that patients had cancer pain if they complained of cancer pain (NRS≥1), regardless of treatment. Patients with pain more than VAS 3 were usually considered for treatment in clinical practice. There might be limits to the application of treatment in clinical practice. However, this study is helpful in identifying patients who are expected to have NCP in advance and for monitoring their symptoms and signs by risk factors.” (line 320-325)

Discussion

In the discussion section, the discussion is too long, and the content of the sentence is in conflict. Need to clean up.

1. What is the clinical meaning of this investigation result?

2. P15 199-206, I cannot understand your explanation

3. According to your results, chemotherapy is also a risk factor for NCP. Old age receives less chemotherapy than young age, so why have more NCP in old age? Does cancer pain decrease in young age receiving toxic agent chemotherapy? This section conflicts with your results and needs clarification. In addition, check the reference again.

Response: We would like to thank you for the careful comments. We cleaned up the Discussion section, deleted the sentence, and revised the paragraph. Our study identified age <65 years as risk factors, not old age. We would like to explain the reason that older patients are less likely to receive chemotherapy. 

“In the present study, we analyzed data from a nationwide cancer pain survey in Korea, focusing on predictors of NCP and the clinical significance of patient-reported symptoms and signs most relevant for these predictors in Korean patients. Our study identified five clinical predictors of NCP: age <65 years, a past or current history of chemotherapy, cancer duration of ≥6 months, moderate-to-severe pain (VAS score ≥4), and early-stage cancer (stage I–III). Bulls et al. reported that older adults (≥65 years) with gynecologic cancer are at higher risk for NCP [13]. Other studies have also reported a higher prevalence of neuropathy in older patients [14,15]. In the present study, 20% of the older (≥65 years) and 7% of the younger (<65 years) age groups had not received chemotherapy (Supplementary Table 1). Older patients are less likely to receive chemotherapy, which can explain the increased association of NCP with the younger age group in our study. A recent study reported that the predictors of NCP were associated with the location of neural invasion, a longer disease duration, a higher pain intensity, recent surgery or chemotherapy, the use of one or more adjuvant analgesics, and the presence of an episodic incident or breakthrough pain [16]. This was similar to our study, where NCP was found to be associated with chemotherapy moderate-to-severe pain and longer duration of cancer.” (line 213--240)

4. P16 L217-239 Combine in one paragraph.

Delete "Failure of identifying NCP leads to unsatisfactory pain management in Asian countries where physician–patient communication tends to be reserved or of short duration"(??)

Response: We would like to thank you for the careful comments. As you advised, we revised the manuscript as follows. 

“Early neuropathic symptoms and signs of platinum analog-treated patients include loss of ankle jerks and decreased vibratory sensibility in the toes associated with numbness, tingling or paresthesias in finger and toes and prolonged treatment may worsen those symptoms. Pin and temperature sensation, joint position and light touch perception are less severely affected [19, 20]. In our study, the most frequently reported symptoms were tingling, electric shock, and pins and needles, and these symptoms were most frequently reported by both NCP and non-NCP groups. It is possible for neuropathic pain to begin with common symptoms such as tingling, pins and needles and electric shocks. A Spanish prospective study, conducted over 1 month, that recruited 366 patients with cancer, aged ≥18 years, diagnosed with NCP (DN4 ≥4), and with moderate-to-severe pain (VAS ≥4), reported that the most common symptoms were tingling (79.8%), pins and needles (76.2%) and electric shocks (72.4%) [21]. This showed a similar distribution of descriptive symptoms as the NCP group in the present study. In contrast, an Indian study reported that a pricking type of pain was the most characteristic feature (47.8%), followed by shooting pain (38.3%), in patients with NCP [22]. These differences in descriptive symptoms might be associated with differences in study design, ethnicity, or expressive language used to express patients’ pain. The symptoms and signs of highest relevance to NCP were prick hypoesthesia and touch hypoesthesia. Further studies to clarify symptom definitions and to elucidate the pathogenesis of NCP are needed for its effective management.” (line 255-273)

---

## [Decision Letter · Decision Letter 1]

24 May 2021

Significance of descriptive symptoms and signs and clinical parameters as predictors of neuropathic cancer pain

PONE-D-20-33583R1

Dear Dr. Shin,

We’re pleased to inform you that your manuscript has been judged scientifically suitable for publication and will be formally accepted for publication once it meets all outstanding technical requirements.

Kind regards,

Ahmed Negida, MD

Academic Editor

PLOS ONE

Additional Editor Comments (optional):

Reviewers' comments:

Reviewer's Responses to Questions

**Comments to the Author**

1. If the authors have adequately addressed your comments raised in a previous round of review and you feel that this manuscript is now acceptable for publication, you may indicate that here to bypass the “Comments to the Author” section, enter your conflict of interest statement in the “Confidential to Editor” section, and submit your "Accept" recommendation.

Reviewer #2: All comments have been addressed

Reviewer #3: All comments have been addressed

2. Is the manuscript technically sound, and do the data support the conclusions?

Reviewer #2: Yes

Reviewer #3: Yes

3. Has the statistical analysis been performed appropriately and rigorously? 

Reviewer #2: Yes

Reviewer #3: Yes

4. Have the authors made all data underlying the findings in their manuscript fully available?

Reviewer #2: Yes

Reviewer #3: No

5. Is the manuscript presented in an intelligible fashion and written in standard English?

Reviewer #2: Yes

Reviewer #3: Yes

6. Review Comments to the Author

Reviewer #2: (No Response)

Reviewer #3: (No Response)

7. PLOS authors have the option to publish the peer review history of their article (what does this mean?). If published, this will include your full peer review and any attached files.

Reviewer #2: No

Reviewer #3: No

---

## [Editor Report · Acceptance letter]

5 Aug 2021

PONE-D-20-33583R1 

Significance of descriptive symptoms and signs and clinical parameters as predictors of neuropathic cancer pain 

Dear Dr. Shin:

I'm pleased to inform you that your manuscript has been deemed suitable for publication in PLOS ONE. Congratulations! Your manuscript is now with our production department. 

Kind regards, 

on behalf of

Dr. Ahmed Negida 

Academic Editor

PLOS ONE